# The Past, Present, and Future of Typological Databases in NLP

**Emi Baylor***
McGill University
Mila Quebec AI Institute
emily.baylor@mail.mcgill.ca

**Esther Ploeger***
Dept. of Computer Science
Aalborg University
espl@cs.aau.dk

**Johannes Bjerva***
Dept. of Computer Science
Aalborg University
jbjerva@cs.aau.dk

## Abstract

Typological information has the potential to be beneficial in the development of NLP models, particularly for low-resource languages. Unfortunately, current large-scale typological databases, notably WALS and Grambank, are inconsistent both with each other and with other sources of typological information, such as linguistic grammars. Some of these inconsistencies stem from coding errors or linguistic variation, but many of the disagreements are due to the discrete categorical nature of these databases. We shed light on this issue by systematically exploring disagreements across typological databases and resources, and their uses in NLP, covering the past and present. We next investigate the future of such work, offering an argument that a continuous view of typological features is clearly beneficial, echoing recommendations from linguistics. We propose that such a view of typology has significant potential in the future, including in language modeling in low-resource scenarios.

## 1 Introduction

Linguistic typology is concerned with investigating languages based on their properties, including similarities and differences in grammar and meaning (Croft, 2002). This type of systematic investigation of language allows for comparing and contrasting languages, providing insights into language, human cognition, and allowing for computational linguistic investigations into language structures. Furthermore, typological properties offer a promising avenue to enabling truly low-resource language technology — efficiently using what we know *about* languages may be the key to effective transfer learning, allowing language models to be as impactful for smaller communities as they are for the global elite.

Typological information is meticulously organised in resources such as the World Atlas of Language Structures (WALS) and Grambank. The information in these databases is typically gathered from other resources, e.g., provided by field linguists in the form of grammars. However, these databases are severely limited by two factors: (i) they contain significant disagreements with one another, implying classification errors, and denoting a hurdle to investigation of language variation across languages and continents; and (ii) they explicitly enforce a categorical view of language, where a continuous view may be more appropriate.

In this paper, we take a long-term view on typology in the context of computational linguistics, NLP, and large language models. We outline past work in the intersection of linguistic typology and NLP, lay out the current state of typological databases, and provide empirical evidence of their inconsistencies. Finally, we take a look at the potential future of such work, highlighting cases where a *continuous* view of typological features is clearly appropriate. The core of our work is to argue for the awareness of these limitations in the field, and to encourage researchers to focus on the continuous scale view of typology.

## 2 Typological Resources

### 2.1 WALS

The World Atlas of Language Structures (WALS) is a large knowledge base of typological properties at the lexical, phonological, syntactic and semantic level (Dryer and Haspelmath, 2013). The data in WALS is based on descriptions of linguistic structure from a wide variety of academic works, ranging from field linguistics to grammars describing the nuances of individual grammatical uses. Typically, WALS categorisations of features are expressed as absolute features, disallowing features with conflicting values. For instance, languages

---

*\* All authors contributed equally to this work.*

are described as being strictly SVO word order, or as strictly having a certain number of vowels. We argue that this view is limited.

## 2.2 Grambank

Unlike WALS, Grambank focuses on what is possible in a language, instead of what is most common. For example, WALS features 81A and 81B present the one or two most dominant subject, object, and verb word orders (Dryer, 2013). Grambank, on the other hand, uses features 131, 132, and 133 to describe the possible word orders, with each feature using binary values to indicate whether or not their given word order is present in the language (Skirgård et al., 2023).

## 2.3 Wikipedia as a Typological Resource

Wikipedia has potential as a typological resource due to its broad and in-depth linguistic coverage. There are thousands of articles detailing varying aspects of different languages, with the articles themselves written in a variety of different languages. In this paper, we focus solely on Wikipedia articles written in English.

Most of the other resources discussed here are created by trained linguists. While the requirement of formal qualifications can increase the reliability of the information presented, it does limit the number of people who can contribute to the resources. Wikipedia, which has a much lower barrier to contribution, allows a much broader group of people to share their linguistic knowledge. As such, it has the potential to contain relevant user-centred information that other typological resources lack.

## 3 Quantifying the Mismatches in Typological Databases

It is well-established that resources such as WALS contain errors. For instance, Plank (2009) carry out an in-depth investigation of the features described for German, noting that 'a non-negligible proportion of the values assigned is found to be problematic, in the sense of being arbitrary or uncertain in view of analytic alternatives, unappreciative of dialectal variation, unclear as to what has been coded, or factually erroneous' (Plank, 2009). We begin by building upon this work, attempting to identify further errors and problems in WALS, by contrasting it with another existing database, Grambank. This allows us to use a semi-automatic approach to quantify errors in typological databases. We match

|             | WALS | Grambank                    |
|-------------|------|-----------------------------|
| **Exact Match** | VSO  | verb-initial only           |
| **Soft Match**  | VSO  | verb-initial and verb-final |
| **No Match**    | VSO  | verb-medial and verb-final  |

Table 1: Examples of cases of exact match, soft match, and no match.

a subset of features from each database to one another. This is straightforward in many cases, as some features have exact counterparts in the other database, and can therefore be easily compared. Other features do not have exact counterparts.

As an example, feature 81A in WALS uses seven different labels (SOV, SVO, VSO, VOS, OVS, OSV, and 'No dominant order') to represent the most dominant word order in a given language. Feature 81B is similar, but uses five different labels ('SOV or SVO', 'VSO or VOS', 'SVO or VSO', 'SVO or VOS', and 'SOV or OVS') to represent the two most dominant word orders for languages with two dominant word orders. Comparatively, the word order features in Grambank (131, 132, and 133) have binary labels which represent whether or not verb-initial, verb-medial, or verb-final word orders are present in the language. Given these differences, we use two methods to compare the word orders indicated in each database. In the first, we check whether or not the labels in WALS exactly match the labels in Grambank (exact match). In the second, we take into account the fact that WALS represents only the dominant word order(s) in a language, while Grambank represents all word orders that occur in that language (soft match). To do so, we count as matches the languages where the WALS labels are a subset of the Grambank labels. Examples of exact match, soft match, and no match cases can be found in Table 1.

The results of this initial experiment, displayed in Table 2, show that agreement between these two established typological databases is often quite low. For example, the agreement on noun-adjective ordering is as low as 72.63% in the Eurasia Macroarea, 57.14% in the North America Macroarea, and 80.13% on average across all languages studied.

The main inconsistencies and errors in WALS and Grambank fall into one of three categories: factual errors, differences due to differing language varieties, or simplifications introduced by the discretization of data to fit the database formats.

| Macroarea | W-Order | W-Order (soft) | Noun/Adj. Order | Num./Noun Order | Assoc. Plural | Redupl. | Mean |
|---|---|---|---|---|---|---|---|
| Africa | 78.38% | 95.95% | 96.61% | 98.11% | 100.0% | 0.00% | 78.17% |
| Australia | 0.00% | 100.0% | | 100.0% | | 0.00% | 50.0% |
| Eurasia | 61.43% | 98.10% | 72.63% | 95.05% | 66.67% | 35.48% | 71.56% |
| North America | 80.30% | 100.00% | 57.14% | 71.43% | 66.67% | 23.53% | 66.51% |
| Papunesia | 72.60% | 95.89% | 92.98% | 87.93% | 0.00% | 0.00% | 58.23% |
| South America | 7.84% | 100.0% | 66.67% | 97.62% | | 50.00% | 64.43% |
| Average | 58.12% | 98.04% | 80.13% | 93.31% | 64.62% | 20.0% | 69.04% |

Table 2: Agreement between a subset of WALS and Grambank feature encodings, aggregated by language macroarea.

## 3.1 Factual Errors

Factual errors can be either due to an error in the source grammar, or an error made when encoding the data from the source into the database. Errors originating in the source grammar can be difficult to identify and fix, because outside, independent information is needed. Errors made in the encoding process are easier to locate and rectify, by comparing the database to the source material. That being said, this process can still be time-consuming.

## 3.2 Linguistic Variations

Inconsistencies between data sources are not necessarily factual errors. Language changes over time, and varies across speakers and domains. Inconsistencies can occur when one source describes language from one time, domain, and speaker group, while another describes language from a different time, domain, or speaker group. In these cases, the sources are accurately describing the language, but are creating inconsistencies by under-specifying the context of the language they are describing.

We investigate this variation in language use across the Universal Dependencies treebanks for a small set of languages: Dutch, English, French, and Japanese. These languages represent the two dominant Noun-Adjective feature orderings, with French notably having a large degree of variation depending on the lexical items involved. Table 3 shows the preference of Noun-Adjective ordering, based on a count of dependency links. Concretely, we investigate the head of any adjective's *amod* dependency, and investigate whether this is before or after the adjective itself. Interestingly, French has a substantial difference across linguistic variations, with spoken language having a slight preference for A-N ordering, as compared to the medical domain containing mostly N-A ordering.

Conspicuously, the first English treebank in our sample contains about 31% Noun-Adjective orderings, while this is only 1% or 2% for the other

1. Which flights leave Chicago on April twelfth and arrive in Indianapolis in the morning?

2. All flights from Washington DC to San Francisco after 5 pm on November twelfth economy class.

3. Round trip air fares from Pittsburgh to Philadelphia less than 1000 dollars.

4. What is the ground transportation available in Fort Worth Texas?

Figure 1: Example sentences with Noun-**Adjective** orderings taken from the Universal Dependencies English Atis treebank.

English treebanks. This is explained by the fact that many of these Noun-Adjective orderings are dates, examples of which can be seen in sentences 1 and 2 of Figure 1. The rest are either sentences written in the style of news headlines, as in sentence 3 in Figure 1, or sentences containing the word *available* as an adjective, as in sentence 4 in Figure 1.

## 3.3 Discretization of Data

The current formats of WALS and Grambank necessitate the division of linguistic features into fairly discrete categories. Even when some variation is included, it is discretized. For example, WALS feature 81A has a label "No dominant order", which allows for the capture of languages that do not have one main way of ordering subjects, objects, and verbs. This puts a language that alternates between SVO, SOV, and VSO only in the same category as a language that has completely free word order, erasing any information on the fundamental ways that these languages differ. Language does not fit into discrete categories, and forcing it to do so in these databases removes crucial information (Levshina et al., 2023). This is well-established in linguistics,

| Language | Domain | N-A | A-N |
|---|---|---|---|
| **Dutch** | news | 1% | 99% |
| | wiki | 1% | 99% |
| **English** | news, nonfiction | 31% | 69% |
| | news, wiki | 1% | 99% |
| | blog, social | 1% | 99% |
| | blog, email, reviews, social, web | 2% | 98% |
| | legal, news, wiki | 2% | 98% |
| | spoken | 1% | 99% |
| **French** | news, nonfiction | 67% | 33% |
| | news, wiki | 61% | 39% |
| | spoken | 44% | 56% |
| | legal, news, wiki | 67% | 33% |
| | spoken | 34% | 66% |
| | medical, news, nonfiction, wiki | 72% | 28% |
| | blog, news, reviews, wiki | 66% | 34% |
| **Japanese** | blog, fiction, news, nonfiction, wiki | 13% | 87% |
| | blog, news | 5% | 95% |
| | news, wiki | 0% | 100% |

Table 3: Variation in noun-adjective order in UD treebanks across domains.

and has also been pointed out as a potential future avenue for research in computational linguistics and NLP (Bjerva and Augenstein, 2018; Bjerva et al., 2019a). This is further corroborated by our results in Table 3, where even languages with strict A-N ordering such as Dutch, English, and Japanese show some variety.

Discretization in databases often leads to inconsistencies between these categorical databases and written sources that can be more nuanced. For example, WALS classifies Hungarian as having "No dominant order" of subjects, objects, and verbs. The Wikipedia article about Hungarian, however, states "The neutral word order is subject–verb–object (SVO). However, Hungarian is a topic-prominent language, and so has a word order that depends not only on syntax but also on the topic–comment structure of the sentence". [1]

## 4 Typological Databases and Large Language Models

These data issues make it difficult to successfully incorporate typological features into language models. As a simplified example, we attempt to fine-tune large language models (LLMs) to predict a variety of languages' WALS features from Wikipedia articles about these languages. To do so, we extract the sections in these Wikipedia articles related to word ordering, using string matching to verify that they contain sufficient word order-related informa-

[1] https://en.wikipedia.org/wiki/Hungarian_language, accessed 16 June 2023.

tion. We then fine-tune pretrained LLMs, with an added classification head, to predict a language's WALS word order feature, given that language's Wikipedia snippet as input.

As hypothesised based on the above data issues, the fine-tuned models generally do not perform well. More often than not, the models output the same predictions as a baseline model that predicts the majority class for the relevant WALS feature, implying the models are able to extract little to none of the input typological information. These results can be found in Table 4.

| Model | Accuracy |
|---|---|
| **Baseline (majority class)** | 52.16% |
| **bert-base-uncased** | 59.05% |
| **roberta-base** | 52.16% |
| **xlm (xlm-mlm-en-2048)** | 56.11% |

Table 4: Large language model results when fine-tuned to classify English Wikipedia articles about languages into their WALS word order feature class (feature 81A).

## 5 The Past, Present, and Future of Typology in NLP

In spite of the inconsistencies and errors we remark upon in typological databases such as WALS, there is a body of research in NLP which largely appears to be unaware of these issues.

**Past** Considering the past years of research, typological databases have mainly been used in the context of feature predictions. Methodologically speaking, features are typically predicted in the context of other features, and other languages (Daumé III and Campbell, 2007; Teh et al., 2009; Berzak et al., 2014; Malaviya et al., 2018; Bjerva et al., 2019c,a, 2020, 2019b; Vastl et al., 2020; Jäger, 2020; Choudhary, 2020; Gutkin and Sproat, 2020; Kumar et al., 2020). That is to say, given a language $l \in L$, where $L$ is the set of all languages contained in a specific database, and the features of that language $F_l$, the setup is typically to attempt to predict some subset of features $f \subset F_l$, based on the remaining features $F_l \setminus f$. Typically, this language may be (partially) held out during training, such that a typological feature prediction model is fine-tuned on $L \setminus l$, before being evaluated on language $l$. Variations of this setup exist, with attempts to control for language relatedness in training/test sets, using genealogical, areal, or structural similarities (Bjerva

et al., 2020; Östling and Kurfalı, 2023).

**Present**   A more recent trend is using typological feature prediction to interpret what linguistic information is captured in language representations of, for example, a neural language model (Malaviya et al., 2017; Choenni and Shutova, 2022; Stanczak et al., 2022; Östling and Kurfalı, 2023; Bjerva, 2023). Such work leans on the intuition that, if a model encapsulates typological properties, it should be possible to accurately predict these based on the model's internal representations.

Given the inconsistencies and errors in typological databases, it is difficult to firmly establish what may be learned about either language or about NLP models by this line of research.

**Future**   Recent work in linguistics has argued for a gradient approach to word order, or in other words, a continuous scale view on this type of typological features (Levshina et al., 2023). We echo this recommendation, corroborated by the fact that languages typically lie on a continuum in terms of word orders. Furthermore, we argue that the lack of this view is the root cause of some of the disagreements between established linguistic resources. We provide the added perspective that taking this continuous view will also potentially lead to improvements in incorporating typological features in NLP models. Based on this line of argumentation, it is perhaps not so surprising that to date, work on incorporating typological features in NLP models has only led to limited effects on performance (Ponti et al., 2019; Üstün et al., 2022).

## 6   Conclusion

Typological information has the potential to improve downstream NLP tasks, particularly for low-resource languages. However, this can only happen if the typological data we have access to is reliable. We argue that a continuous view of typological features is necessary in order for this to be the case, and for typology to truly have an impact in low-resource NLP and language modelling. Until this is amended, we argue that it is important that the community is aware of these inherent limitations when using typology for interpretability of models.

## Limitations

In our investigation into mismatches between typological resources, we only investigate a subset.

While unlikely, it is entirely possible that the remaining features in the databases match. Furthermore, the dependency count experiments for Noun-Adjective ordering assume that annotations in UD are correct.

## Ethics Statement

No human data was gathered in this work. As this paper focuses on a recommendation for future work in NLP to be aware of a core limitation in typological databases, we do not foresee any significant ethical issues stemming from this line of research.

## Acknowledgements

This work was supported by a *Semper Ardens: Accelerate* research grant (CF21-0454) from the Carlsberg Foundation. EB was further supported by the McGill University Graduate Mobility Award to travel to AAU to carry out this work.

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
