# OpenReview forum: "The Past, Present, and Future of Typological Databases in NLP"
_EMNLP/2023/Conference — EMNLP 2023 Findings_

### Official Review · Reviewer_wc1P · 2023-07-27

**Soundness:** 2

**Excitement:**

2: Mediocre: This paper makes marginal contributions (vs non-contemporaneous work), so I would rather not see it in the conference.

**Paper Topic And Main Contributions:**

The paper focuses on typological databases. It discusses, in particular, their limits in terms of coverage and flexibility for representing typological properties of languages. To this aim, the authors compare WALS and Grambank to demonstrate that these resources frequently offer contrasting evidence. Furthermore, the authors show how Wikipedia can be used to acquire fine-grained information about linguistic variation within a language and briefly show an experiment where this information is used to fine-tune LLMs.

**Reasons To Accept:**

- The paper is well written and the topic it addresses is interesting. It is worth discussing how to integrate existing linguistic resources with LLMs.
- In general, the paper has no specific critical issues worth reporting.

**Reasons To Reject:**

- Despite lacking any specific issues, the paper is found to be lacking in originality. The topics related to typological databases discussed within the paper, namely, inconsistency across databases and the inflexibility to represent gradients of frequencies of phenomena across time and domain, have already been explored in previous works. Some of these relevant works are cited in the bibliography of this paper.
- The metrics proposed to account for matches and mismatches are rather trivial and were only tested for a single pair of databases.
- Section 4 presents the experiments with a degree of vagueness, and there is no in-depth discussion about the results.
- The paper's ultimate goal, which is to shed light on the limitations of typological databases when interpreting NLP models, is not fully achieved, as the reported evidence does not introduce any novel insights.

**Reproducibility:**

2: Would be hard pressed to reproduce the results. The contribution depends on data that are simply not available outside the author's institution or consortium; not enough details are provided.

**Reviewer Confidence:**

4: Quite sure. I tried to check the important points carefully. It's unlikely, though conceivable, that I missed something that should affect my ratings.

---

> ### Author Rebuttal · Authors · 2023-08-29
>
> Thank you for reading our paper and providing your assessment of our submission and. Below, we address each of your concerns:
>
>
> > Lack of novelty
>
> While we indeed explore areas similar to those in previous works, and indeed draw conclusions that mirror conclusions that other authors have drawn, we argue that our contribution is valuable regardless, as the mentioned issues have not been considered specifically from the perspective of NLP before. While our findings are not novel, the paper is novel because we investigate the claims with new data (Grambank, Wikipedia) and provide a classification of the issues. We argue that it is worth presenting this new supporting evidence, even if the conclusions drawn from it are not wholly novel. We will emphasize this point in the camera-ready version.
>
> > Triviality of metrics
>
> While the metrics used to account for matches and mismatches are indeed trivial, a more complex methodology was not required for this quantification, and was therefore not used. We refer the reviewer to the review policy, which states that ‘The goal is to solve the problem, not to solve it in a complex way. Simpler solutions are in fact preferable, as they are less brittle and easier to deploy in real-world settings.’
> Our contribution serves to empirically establish mismatches in these databases, and opens up for exploration into more complex methodology down the line, when appropriate - e.g.  if necessary for other typological features.
>
> > Single pair of databases
>
> While it is true that we only tested for a single pair of databases, these are two of the most frequently used typological databases in NLP. While they do not necessarily represent the broad range of all typological databases, their degree of mismatch implies that much of typology-based NLP research is based on data that is, at the very least, inconsistent. We will make this point more explicit in the camera-ready version.

---

### Official Review · Reviewer_mEdM · 2023-08-03

**Soundness:** 2

**Excitement:**

2: Mediocre: This paper makes marginal contributions (vs non-contemporaneous work), so I would rather not see it in the conference.

**Paper Topic And Main Contributions:**

The article discusses the role of linguistic typology in investigating languages based on their properties and the potential implications for NLP and LLMs. It emphasizes the need to understand and address the limitations and inconsistencies in typological databases such as the World Atlas of Language Structures or Grambank. These databases suffer from classification errors, disagreements, and a categorical view of language features. The article advocates for a continuous view of typological features, considering the nuances and variations in language structures.

The authors present empirical evidence of the inconsistencies in typological databases by comparing data from WALS and Grambank, and they categorize the errors into factual errors, linguistic variations, and simplifications due to data discretization. They show that these issues hinder the successful incorporation of typological features into language models.

The article also highlights the past, present, and future trends in incorporating typological features in NLP, discussing research that uses these features for prediction and interpretation in language models.

**Reasons To Accept:**

1. The paper addresses a relevant and important topic by discussing the potential impact of linguistic typology on NLP and large language models.

2. The paper clearly articulates the problem it aims to address – the inconsistencies and errors in typological databases and their impact on NLP tasks.

3. The paper presents empirical evidence of the inconsistencies in typological databases by comparing data from different sources.

4. The authors categorize the sources of errors and limitations, classifying them into factual errors, linguistic variations, and data discretization issues.


**Reasons To Reject:**

1. While the paper identifies the limitations and challenges of using typological databases, it provides relatively limited discussion about potential solutions to address these issues. While the authors suggest a shift towards a continuous view of typological features, they could have explored more concrete strategies or approaches that researchers could adopt to mitigate the inconsistencies and errors in the databases.

2. The paper acknowledges that it only investigates a subset of typological features for certain languages, which raises questions about the generalizability of the findings.

3. The paper could benefit from a more detailed comparative analysis of the specific discrepancies between different databases.

4. The paper mentions that fine-tuning large language models to predict typological features from Wikipedia articles yields limited results. However, it lacks a detailed discussion of the potential implications of these results on the actual performance of NLP models in downstream tasks.

5. The discussion of linguistic variations in the paper is limited to a few languages and domains. A more comprehensive analysis across a broader range of languages and contexts would provide a more nuanced understanding of how linguistic variations contribute to the inconsistencies in typological databases.

6. The conclusions might not be entirely novel for experts in the field. The paper could have gone further in proposing innovative approaches or avenues for future research beyond advocating for a continuous view of typological features.

**Reproducibility:**

3: Could reproduce the results with some difficulty. The settings of parameters are underspecified or subjectively determined; the training/evaluation data are not widely available.

**Reviewer Confidence:**

2: Willing to defend my evaluation, but it is fairly likely that I missed some details, didn't understand some central points, or can't be sure about the novelty of the work.

**Typos Grammar Style And Presentation Improvements:**

Typos and other minor issues:

– 025: languages languages > languages

– 093: has potential > has the potential

– Table 1 caption: Examples cases > Examples ofcases?

– font in Table 2 is really small, there must be some way to reformet it (e.g. by adding a second row to the header)

– 245: where a language model was provided the Wikipedia > where a language model was provided with the Wikipedia

– 288: leans on the the intuition > leans on the intuition

– 422: 'Why we need a gradient approach...' already has a proper citation (e.g. with page numbers) so please correct.

In the bibliography: please replace arXiv entries with references to published papers (e.g.: Ostling and Kurfalı's "Language embeddings sometimes contain typological generalizations" is now available at https://direct.mit.edu/coli/article/doi/10.1162/coli_a_00491/116637)

---

> ### Author Rebuttal · Authors · 2023-08-29
>
> Thank you for reading our paper and providing your assessment of our submission. We appreciate that you find our problem formulation, empirical evidence and categorization of causes valuable contributions. Below, we address each of your concerns (reasons to reject):
>
> > Lack of proposed solutions and novelty (reason 1 and 6)
>
> While we agree that constructing concrete solutions to the problems we highlight is a promising future research direction, this is beyond the goal of our paper. We argue that our paper contains valuable contributions, as the mentioned issues have not been considered specifically from the perspective of NLP before. While our findings are not novel, the paper is novel because we investigate the claims with new data (Grambank, Wikipedia) and provide a classification of the issues. Our paper therefore provides a starting point for future work in solving these issues, which we will emphasize in the camera-ready version.
>
> > Limited scope (reason 2 and 5)
>
> While we acknowledge that experimental results with more languages, domains and features (we are unclear what the reviewer refers to with ‘contexts’) would be interesting, we argue that our experiments suffice to highlight the central issues on this topic. Additionally, adding more features would introduce several new issues: features with lower coverage may be skewed towards certain (Indo-European) languages, while it is important for our conclusions that we take a broad range of languages into account. We will add this note into our paper.
>
> > Lack of detailed comparison (reason 3)
>
> It is unclear to us what the reviewer means with ‘detailed comparative analysis’. The purpose of our experiments is to support the claims of our position paper, and that our experiments provide sufficient details for that (i.e. results listed per feature, soft vs. exact word order matching, results split per macroarea in Table 2). We are curious what other details the reviewer deems relevant to add.
>
> > Implications on downstream tasks (reason 4)
>
> The prediction task in this paper tests, in a straightforward way, whether or not the typological information at hand can be learned from language descriptions by deep learning models. If it had worked, we would then move on towards incorporating the information contained in the Wikipedia articles into more directly useful downstream tasks. For example, we would potentially attempt to use the information to improve low-resource language models, testing whether or not the direct linguistic information could make up for some of the lack of raw training data. That being said, the straightforward prediction task failed, indicating that the information (in conjunction with typological databases) may not be reliable enough to improve these other downstream tasks.

---

### Official Review · Reviewer_PNkm · 2023-08-18

**Soundness:** 4

**Excitement:**

3: Ambivalent: It has merits (e.g., it reports state-of-the-art results, the idea is nice), but there are key weaknesses (e.g., it describes incremental work), and it can significantly benefit from another round of revision. However, I won't object to accepting it if my co-reviewers champion it.

**Missing References:**

I wouldn't say missing but these two are an interesting and a related read

Survey on the Use of Typological Information in Natural Language Processing Helen O'Horan, Yevgeni Berzak, Ivan Vulić, Roi Reichart, Anna Korhonen COLING 2016

Linguistically Na ̈ıve != Language Independent: Why NLP Needs Linguistic Typology Emily M. Bender EACL 2009 Workshop on the Interaction between Linguistics and Computational Linguistics

**Paper Topic And Main Contributions:**

In this paper the authors study large scale typological databases such as WALS and Grambank and address the issue of inconsistent typological information among each other. The authors argue that the disagreements between the databases arise due to the categorical nature of the databases Furthermore, they point that this enforced categorical view of the language proves to be a major limitation for these databases. The authors suggest future work to focus on a continuous scale view of typology.

The main contributions in this paper are:
The authors explore Wikipedia as a potential resource for typological information.
They contrast the two databases WALS and Grambank, to identify the errors in the dataset. For this they consider the subset of the features from one database to the other and find that the main inconsistencies in WALS and Grambank fall into three categories a) factual errors, b) differing language variety and c) discretization of data and provide examples to support their claims.
Finally the authors experiment with a set of fine tuned BERT based models to predict WALS features of languages from Wikipedia articles and conclude that these large language models perform poorly, matching the Accuracy of their simple baseline

**Reasons To Accept:**

This paper brings into light the typological inconsistencies and the errors that exist in the various large scale databases such as WALS and Grambank. They compare and contrast these two databases and quantify the errors that exist in them. This is essential because ensuring the accuracy and the reliability of data in these datasets is crucial for conducting credible research especially when cross-linguistic studies is done.
This paper also encourages future research to adopt a more continuous scale of typology than the existing enforced category nature of it. This could potentially solve some of the existing discrepancies in these large-scale typological databases.
The paper is also well structured with very clear and easy to understand language.

**Reasons To Reject:**

Although the paper has good motivation, the experimental section falls short by a little bit. The authors report that the transformer based models couldn't perform well in the classification task on WALS features of a set of languages. Did the authors consider using any other class of language models like generative models with prompt engineering in their approach ? Also, addressing limitations mentioned in the paper itself could make for a stronger research

**Reproducibility:**

4: Could mostly reproduce the results, but there may be some variation because of sample variance or minor variations in their interpretation of the protocol or method.

**Reviewer Confidence:**

4: Quite sure. I tried to check the important points carefully. It's unlikely, though conceivable, that I missed something that should affect my ratings.

**Typos Grammar Style And Presentation Improvements:**

Line number 25 reads  "Linguistic typology is concerned with investigating languages languages based on their properties..."
Line number 152-154: "we count as matches the languages where... " The sentence construction seems a bit confusing. Please ignore if it was intended to be written that way.

---

> ### Author Rebuttal · Authors · 2023-08-29
>
> Thank you for providing your feedback on our paper. We appreciate that you value the contributions of our paper and found our findings useful. Below, we address your main concern.
>
> > Experiments with other language models
>
> We considered attempting the training with other types of deep learning models, however it became clear when analyzing the models’ errors that the source and target data were too inconsistent to be learned without memorization. We had previously trained on simpler models, such as basic bag-of-word models, but, as would be expected in light of the data issues, these performed poorly.
>
> > Reproducibility score
>
> To address reproducibility concerns, we will now include detailed information to better describe our LLM training process. We will also be releasing the training data used in these experiments, as well as a repository of code needed to reproduce the experiments.

---

### Meta-Review · Area_Chair_UZxu · 2023-09-19

**Recommendation:** 3

**Metareview:**

Reviewers of this paper found the topic to be relevant and clear, and especially found that the problems pointed out in past approaches were valid. Generally, the reviewers didn’t seem to point to many specific methodological issues that should result in lower soundness scores but found the lack of discussion in many areas to leave the paper feeling a bit shallow, noting that it could benefit from another round of revision to expand these areas. It was also noted that the scope was narrow in terms of languages and features studied, but given this is a short paper submission, I don’t think it is a major problem given that this did not seem to invalidate any of the findings in the eyes of the reviewers. Reviewers would have liked to have seen more proposed solutions to the issues raised, and in general, were not overly excited about the paper.

---

### Decision · Program_Chairs · 2023-10-07

**Decision:**

Accept-Findings

**Comment:**

Reviewers of this paper found the topic to be relevant and clear, and especially found that the problems pointed out in past approaches were valid. Generally, the reviewers didn’t seem to point to many specific methodological issues that should result in lower soundness scores but found the lack of discussion in many areas to leave the paper feeling a bit shallow, noting that it could benefit from another round of revision to expand these areas. It was also noted that the scope was narrow in terms of languages and features studied, but given this is a short paper submission, I don’t think it is a major problem given that this did not seem to invalidate any of the findings in the eyes of the reviewers. Reviewers would have liked to have seen more proposed solutions to the issues raised, and in general, were not overly excited about the paper.